# Learning Directed Graphical Models with Optimal Transport

## Abstract

Estimating the parameters of a probabilistic directed graphical model from incomplete data remains a long-standing challenge. This is because, in the presence of latent variables, both the likelihood function and posterior distribution are intractable without further assumptions about structural dependencies or model classes. While existing learning methods are fundamentally based on likelihood maximization, here we offer a new view of the parameter learning problem through the lens of optimal transport. This perspective licenses a framework that operates on many directed graphs without making unrealistic assumptions on the posterior over the latent variables or resorting to black-box variational approximations. We develop a theoretical framework and support it with extensive empirical evidence demonstrating the flexibility and versatility of our approach. Across experiments, we show that not only can our method recover the ground-truth parameters but it also performs competitively on downstream applications, notably the non-trivial task of discrete representation learning.

## 1  Introduction

Learning probabilistic directed graphical models (DGMs, also known as Bayesian networks) with latent variables is an important ongoing challenge in machine learning and statistics. This paper focuses on parameter learning, i.e., estimating the parameters of a DGM given its known structure. Learning DGMs has a long history, dating back to classical indirect likelihood-maximization approaches such as expectation maximization [EM, 15]. However, despite all its success stories, EM is well-known to suffer from local optima issues. More importantly, EM becomes inapplicable when the posterior distribution is intractable, which arises fairly often in practice.

A large family of related methods based on variational inference [VI, 30, 27] have demonstrated tremendous potential in this case, where the evidence lower bound (ELBO) is not only used for posterior approximation but also for point estimation of the model parameters. Such an approach has proved surprisingly effective and robust to overfitting, especially when having a small number of parameters. From a high-level perspective, both EM and VI are based on likelihood maximization in the presence of latent variables, which ultimately requires carrying out expectations over the commonly intractable posterior. In order to address this challenge, a large spectrum of methods have been proposed in the literature and we refer the reader to [5] for an excellent discussion of these approaches. Here we characterize them between two extremes. At one extreme, restrictive assumptions about the structure (e.g., as in mean-field approximations) or the model class (e.g., using conjugate exponential families) must be made to simplify the task. At the other extreme, when no assumptions are made, most existing black-box methods exploit very little information about the structure of the known probabilistic model (for example, in black-box and stochastic variational inference [44, 27], hierarchical approaches [45] and normalizing flows [42]).

Addressing the problem at its core, we hereby propose an alternative strategy to likelihood maximization that does not require the estimation of expectations over the posterior distribution. Concretely, parameter learning is now viewed through the lens of *optimal transport* [54], where the data distribution is the source and the true model distribution is the target. Instead of minimizing a Kullback–Leibler (KL) divergence (which likelihood maximization methods are essentially doing), here we aim to find a point estimate $\theta^*$ that minimizes the Wasserstein distance [WD, 31] between these two distributions.

This perspective allows us to leverage desirable properties of the WD in comparison with other metrics. These properties have motivated the recent surge in generative models, e.g., Wasserstein GANs [1, 9] and Wasserstein Auto-encoders [50]. Indeed, the WD is shown to be well-behaved in situations where standard metrics such as the KL or JS (Jensen-Shannon) divergences are either infinite or undefined [43, 4]. The WD thus characterizes a more meaningful distance, especially when the two distributions reside in low-dimensional manifolds [9]. Ultimately, this novel view enables us to pursue an ambitious goal towards a model-agnostic and scalable learning framework.

**Contributions.** We present an entirely different view that casts parameter estimation as an optimal transport problem [54], where the goal is to find the optimal plan transporting "mass" from the data distribution to the model distribution. To achieve this, our method minimizes the WD between these two distributions. This permits a flexible framework applicable to any type of variable and graphical structure. In summary, we make the following contributions:

- We introduce **OTP-DAG** - an **O**ptimal **T**ransport framework for **P**arameter Learning in **D**irected **A**cyclic **G**raphical models. OTP-DAG is an alternative line of thinking about parameter learning. Diverging from the existing frameworks, the underlying idea is to find the parameter set associated with the distribution that yields the lowest transportation cost from the data distribution.

- We present theoretical developments showing that minimizing the transport cost is equivalent to minimizing the reconstruction error between the observed data and the model generation. This renders a tractable training objective to be solved efficiently with stochastic optimization.

- We provide empirical evidence demonstrating the versatility of our method on various graphical structures. OTP-DAG is shown to successfully recover the ground-truth parameters and achieve competitive performance across a range of downstream applications.

## 2  Background and Related Work

We first introduce the notations and basic concepts used throughout the paper. We reserve bold capital letters (i.e., $\mathbf{G}$) for notations related to graphs. We use calligraphic letters (i.e. $\mathcal{X}$) for spaces, italic capital letters (i.e. $X$) for random variables, and lower case letters (i.e. $x$) for their values.

A **directed graph** $\mathbf{G} = (\mathbf{V}, \mathbf{E})$ consists of a set of nodes $\mathbf{V}$ and an edge set $\mathbf{E} \subseteq \mathbf{V}^2$ of ordered pairs of nodes with $(v, v) \notin \mathbf{E}$ for any $v \in \mathbf{V}$ (one without self-loops). For a pair of nodes $i, j$ with $(i, j) \in \mathbf{E}$, there is an arrow pointing from $i$ to $j$ and we write $i \rightarrow j$. Two nodes $i$ and $j$ are adjacent if either $(i, j) \in \mathbf{E}$ or $(j, i) \in \mathbf{E}$. If there is an arrow from $i$ to $j$ then $i$ is a parent of $j$ and $j$ is a child of $i$. A Bayesian network structure $\mathbf{G} = (\mathbf{V}, \mathbf{E})$ is a **directed acyclic graph** (DAG), in which the nodes represent random variables $X = [X_i]_{i=1}^n$ with index set $\mathbf{V} := \{1, ..., n\}$. Let $\mathrm{PA}_{X_i}$ denote the set of variables associated with parents of node $i$ in $\mathbf{G}$.

In this work, we tackle the classic yet important problem of learning the parameters of a directed graph from *partially observed data*. Let $\mathbf{O} \subseteq \mathbf{V}$ and $X_{\mathbf{O}} = [X_i]_{i \in \mathbf{O}}$ be the set of observed nodes and $\mathbf{H} := \mathbf{V} \backslash \mathbf{O}$ be the set of hidden nodes. Let $P_\theta$ and $P_d$ respectively denote the distribution induced by the graphical model and the empirical one induced by the *complete* (yet unknown) data. Given a fixed graphical structure $\mathbf{G}$ and some set of i.i.d data points, we aim to find the point estimate $\theta^*$ that best fits the observed data $X_{\mathbf{O}}$. The conventional approach is to minimize the KL divergence between the model distribution and the *empirical* data distribution over observed data i.e., $D_{\mathbb{KL}}(P_d(X_{\mathbf{O}}), P_\theta(X_{\mathbf{O}}))$, which is equivalent to maximizing the likelihood $P_\theta(X_{\mathbf{O}})$ w.r.t $\theta$. In the presence of latent variables, the marginal likelihood, given as $P_\theta(X_{\mathbf{O}}) = \int_{X_{\mathbf{H}}} P_\theta(X) dX_{\mathbf{H}}$, is generally intractable. Standard approaches then resort to maximizing a bound on the marginal log-likelihood, known as the evidence lower bound (ELBO), which is essentially the objective of EM [38] and VI [30]. Optimization of the ELBO for parameter learning in practice requires many

considerations. For vanilla EM, the algorithm only works if the true posterior density can be computed exactly. Furthermore, EM is originally a batch algorithm, thereby converging slowly on large datasets [36]. Subsequently, researchers have tried exploring other methods for scalability, including attempts to combine EM with approximate inference [56, 40, 14, 10, 13, 36, 41].

When exact inference is infeasible, a variational approximation is the go-to solution. Along this line, research efforts have concentrated on ensuring tractability of the ELBO via the mean-field assumption [11] and its relaxation known as structured mean field [47]. Scalability has been one of the main challenges facing the early VI formulations since it is a batch algorithm. This has triggered the development of stochastic variational inference (SVI) [27, 26, 16, 29, 8, 7] which applies stochastic optimization to solve VI objectives. Another line of work is collapsed VI that explicitly integrates out certain model parameters or latent variables in an analytic manner [23, 32, 48, 34]. Without a closed form, one could resort to Markov chain Monte Carlo [18, 19, 21], which however tends to be slow. More accurate variational posteriors also exist, namely, through hierarchical variational models [45], implicit posteriors [49, 58, 37, 49], normalizing flows [33], or copula distribution [51]. To avoid computing the ELBO analytically, one can obtain an unbiased gradient estimator using Monte Carlo and re-parameterization tricks [44, 57]. As mentioned in the introduction, an excellent summary of these approaches is discussed in [5, §6]. Extensions of VI to other divergence measures than KL divergence e.g., $\alpha-$divergence or $f-$divergence, also exist [35, 24, 55]. In the causal inference literature, a related direction is to learn both the graphical structure and parameters of the corresponding structural equation model [60, 17]. These frameworks are often limited to additive noise models while assuming no latent confounders.

# 3 Optimal Transport for Learning Directed Graphical Models

We begin by explaining how parameter learning can be reformulated into an optimal transport problem [53] and thereafter introduce our novel theoretical contribution.

We consider a DAG $\mathbf{G}(\mathbf{V}, \mathbf{E})$ over random variables $X = [X_i]_{i=1}^n$ that represents the data generative process of an underlying system. The system consists of $X$ as the set of endogenous variables and $U = \{U_i\}_{i=1}^n$ as the set of exogenous variables representing external factors affecting the system. Associated with every $X_i$ is an exogenous variable $U_i$ whose values are sampled from a prior distribution $P(U)$ independently from other exogenous variables. For the purpose of this work, our framework operates on an extended graph consisting of both endogenous and exogenous nodes (See Figure 1b). In the graph $\mathbf{G}$, $U_i$ is represented by a node with no ancestors that has an outgoing arrow towards node $i$. Consequently, for every endogenous variable, its parent set $\mathrm{PA}_{X_i}$ is extended to include an exogenous variable and possibly some other endogenous variables. Henceforth, every distribution $P_{\theta_i}(X_i|\mathrm{PA}_{X_i})$ can be reparameterized into a deterministic assignment

$$X_i = \psi_i(\mathrm{PA}_{X_i}, U_i), \text{ for } i = 1, ..., n.$$

The ultimate goal is to estimate $\theta = \{\theta_i\}_{i=1}^n$ as the parameters of the set of deterministic functions $\psi = \{\psi_i\}_{i=1}^n$. We will use the notation $\psi_\theta$ to emphasize this connection from now on.

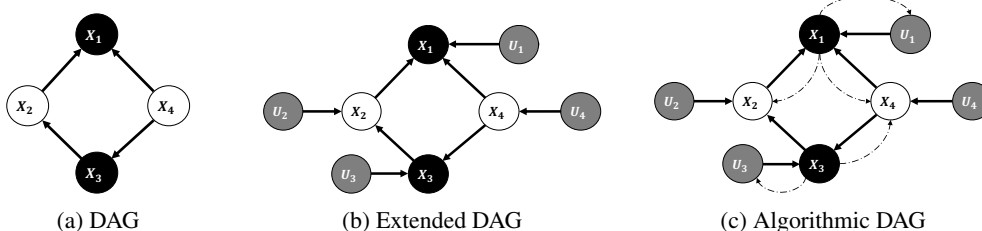

(a) DAG          (b) Extended DAG          (c) Algorithmic DAG

Figure 1: (a) A DAG represents a system of 4 endogenous variables where $X_1, X_3$ are observed (black-shaded) and $X_2, X_4$ are hidden variables (non-shaded). (b): The extended DAG that includes an additional set of independent exogenous variables $U_1, U_2, U_3, U_4$ (grey-shaded) acting on each endogenous variable. $U_1, U_2, U_3, U_4 \sim P(U)$ where $P(U)$ is a prior product distribution. (c) Visualization of our backward-forward algorithm, where the dashed arcs represent the backward maps involved in optimization.

Given the data distribution $P_d(X_\mathbf{O})$ and the model distribution $P_\theta(X_\mathbf{O})$ over the observed set $\mathbf{O}$, the **optimal transport** (OT) goal is to find the parameter set $\theta$ that minimizes the cost of transport between these two distributions. The Kantorovich's formulation of the problem is given by

$$W_c(P_d; P_\theta) := \inf_{\Gamma \sim \mathcal{P}(X \sim P_d, Y \sim P_\theta)} \mathbb{E}_{(X,Y) \sim \Gamma}\big[c(X, Y)\big], \tag{1}$$

where $\mathcal{P}(X \sim P_d, Y \sim P_\theta)$ is a set of all joint distributions of $(P_d; P_\theta)$ and $c : \mathcal{X}_\mathbf{O} \times \mathcal{X}_\mathbf{O} \mapsto \mathcal{R}_+$ is any measurable cost function over $\mathcal{X}_\mathbf{O}$ (i.e., the product space of the spaces of observed variables) that is defined as $c(X_\mathbf{O}, Y_\mathbf{O}) := \sum_{i \in \mathbf{O}} c_i(X_i, Y_i)$ where $c_i$ is a measurable cost function over a space of a certain observed variable.

Let $P_\theta(\mathrm{PA}_{X_i}, U_i)$ denote the joint distribution of $\mathrm{PA}_{X_i}$ and $U_i$ factorized according to the graphical model. Let $\mathcal{U}_i$ denote the space over random variable $U_i$. The key ingredient of our theoretical development is local backward mapping. For every observed node $i \in \mathbf{O}$, we define a stochastic "backward" map $\phi_i : \mathcal{X}_i \mapsto \Pi_{k \in \mathrm{PA}_{X_i}} \mathcal{X}_k \times \mathcal{U}_i$ such that $\phi_i \in \mathfrak{C}(X_i)$ where $\mathfrak{C}(X_i)$ is the constraint set given as

$$\mathfrak{C}(X_i) := \big\{ \phi_i : \phi_i \# P_d(X_i) = P_\theta(\mathrm{PA}_{X_i}, U_i) \big\}.$$

Essentially, $\phi_i$ pushes the data marginal of $X_i$ forward to the model marginal of its parent variables. If $\mathrm{PA}_{X_i}$ are latent variables, $\phi_i$ can be viewed as a stochastic decoder mapping $X_i$ to the conditional density $\phi_i(\mathrm{PA}_{X_i}|X_i)$.

Theorem 1 presents the main theoretical contribution of our paper. Our OT problem is concerned with finding the optimal set of deterministic "forward" maps $\psi_\theta$ and stochastic "backward" maps $\big\{ \phi_i \in \mathfrak{C}(X_i) \big\}_{i \in \mathbf{O}}$ that minimizes the cost of transporting the mass from $P_d$ to $P_\theta$ over $\mathbf{O}$. While the formulation in Eq. (1) is not trainable, we show that the problem is reduced to minimizing the reconstruction error between the data generated from $P_\theta$ and the observed data. To understand how reconstruction works, let us examine Figure 1c. Given $X_1$ and $X_3$ as observed nodes, we sample $X_1 \sim P_d(X_1), X_3 \sim P_d(X_3)$ and evaluate the local densities $\phi_1(\mathrm{PA}_{X_1}|X_1), \phi_3(\mathrm{PA}_{X_3}|X_3)$ where $\mathrm{PA}_{X_1} = \{X_2, X_4, U_1\}$ and $\mathrm{PA}_{X_3} = \{X_4, U_3\}$. The next step is to sample $\mathrm{PA}_{X_1} \sim \phi_1(\mathrm{PA}_{X_1}|X_1)$ and $\mathrm{PA}_{X_3} \sim \phi_3(\mathrm{PA}_{X_3}|X_3)$, which are plugged back to the model $\psi_\theta$ to obtain the reconstructions $\widetilde{X_1} = \psi_{\theta_1}(\mathrm{PA}_{X_1})$ and $\widetilde{X_3} = \psi_{\theta_3}(\mathrm{PA}_{X_3})$. We wish to learn $\theta$ such that $X_1$ and $X_3$ are reconstructed correctly. For a general graphical model, this optimization objective is formalized as

**Theorem 1** *For every $\phi_i$ as defined above and fixed $\psi_\theta$,*

$$W_c\big(P_d(X_\mathbf{O}); P_\theta(X_\mathbf{O})\big) = \inf_{\big[\phi_i \in \mathfrak{C}(X_i)\big]_{i \in \mathbf{O}}} \mathbb{E}_{X_\mathbf{O} \sim P_d(X_\mathbf{O}), \mathrm{PA}_{X_\mathbf{O}} \sim \phi(X_\mathbf{O})} \big[c\big(X_\mathbf{O}, \psi_\theta(\mathrm{PA}_{X_\mathbf{O}})\big)\big], \tag{2}$$

*where* $\mathrm{PA}_{X_\mathbf{O}} := \big[[X_{ij}]_{j \in \mathrm{PA}_{X_i}}\big]_{i \in \mathbf{O}}$.

The proof is provided in Appendix A. It is seen that Theorem 1 set ups a trainable form for our optimization solution. Notice that the quality of the reconstruction hinges on how well the backward maps approximate the true local densities. To ensure approximation fidelity, every backward function $\phi_i$ must satisfy its push-forward constraint defined by $\mathfrak{C}$. In the above example, the backward maps $\phi_i$ and $\phi_3$ must be constructed such that $\phi_1 \# (X_1) = P_\theta(X_2, X_4, U_1)$ and $\phi_3 \# (X_3) = P_\theta(X_4, U_3)$. This gives us a constraint optimization problem, and we relax the constraints by adding a penalty to the above objective.

The **final optimization objective** is therefore given as

$$J_{WS} = \inf_{\psi, \phi} \ \mathbb{E}_{X_\mathbf{O} \sim P_d(X_\mathbf{O}), \mathrm{PA}_{X_\mathbf{O}} \sim \phi(X_\mathbf{O})} \big[c\big(X_\mathbf{O}, \psi_\theta(\mathrm{PA}_{X_\mathbf{O}})\big)\big] + \eta \, D\big(\phi, P_\theta\big), \tag{3}$$

where $D$ is any arbitrary divergence measure and $\eta > 0$ is a trade-off hyper-parameter. $D\big(\phi, P_\theta\big)$ is a short-hand for divergence between all pairs of backward and forward distributions.

This theoretical result provides us with several interesting properties: (1) to minimize the global OT cost between the model distribution and the data distribution, one only needs to characterize the local densities by specifying the backward maps from every observed node to its parents and optimizing them with appropriate cost metrics; (2) all model parameters are optimized simultaneously within a single framework whether the variables are continuous or discrete ; (3) the computational process can be automated without deriving an analytic lower bound or restricting to certain graph-

ical structures. In connection with VI, OTP-DAG is also optimization-based. We in fact leverage modern VI techniques of reparameterization and amortized inference [6] for solving it efficiently via stochastic gradient descent. However, unlike such advances as hierarchical VI, our method does not place any prior over the variational distribution on the latent variables underlying the variational posterior [45]. For providing a guarantee, OTP-DAG relies on the condition that the backward maps are sufficiently expressive to cover the push-forward constraints. We prove further in Appendix A that given a suitably rich family of backward functions, our algorithm OTP-DAG can converge to the ground-truth parameters. Details on our algorithm can be found in Appendix B. In the next section, we illustrate how OTP-DAG algorithm is realized in practical applications.

# 4 Applications

We apply OTP-DAG on 3 widely-used graphical models for a total of 5 different sub-tasks. Here we aim to demonstrate the versatility of OTP-DAG: OTP-DAG can be exploited for various purposes through a single learning procedure. In terms of estimation accuracy, OTP-DAG is capable of recovering the ground-truth parameters while achieving the comparable or better performance level of existing frameworks across downstream tasks.[1]

We consider various directed probabilistic models with either continuous or discrete variables. We begin with (1) Latent Dirichlet Allocation [12] for topic modeling and (2) Hidden Markov Model (HMM) for sequential modeling tasks. We conclude with a more challenging setting: (3) Discrete Representation Learning (Discrete RepL) that cannot simply be solved by EM or MAP (maximum a posteriori). It in fact invokes deep generative modeling via a pioneering development called Vector Quantization Variational Auto-Encoder (VQ-VAE) [52]. We investigate an application of OTP-DAG algorithm to learning discrete representations by grounding it into a parameter learning problem.

Note that our goal is not to achieve the state-of-the-art performance, rather to prove OTP-DAG as a versatile approach for learning parameters of directed graphical models. Figure 2 illustrates the empirical DAG structures of the 3 applications. Unlike the standard visualization where the parameters are considered hidden nodes, our graph separates model parameters from latent variables and only illustrates random variables and their dependencies (except the special setting of Discrete RepL). We also omit the exogenous variables associated with the hidden nodes for visibility, since only those acting on the observed nodes are relevant for computation. There is also a noticeable difference between Figure 2 and Figure 1c: the empirical version does not involve learning the backward maps for the exogenous variables. This stems from an experimental observation that sampling the noise from an appropriate prior distribution at random suffices to yield accurate estimation. We find it to be beneficial in that training complexity can be greatly reduced. In the following, we report the main experimental results, leaving the discussion of the formulation and technicalities in Appendix C. In all tables, we report the average results over 5 random initializations and the best ones are highlighted in bold. In addition, $\uparrow, \downarrow$ indicate higher/lower performance is better, respectively.

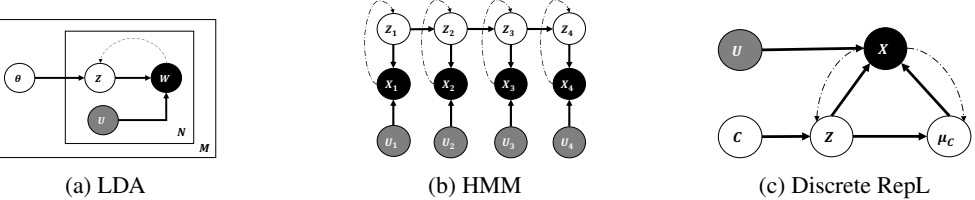

|  (a) LDA | (b) HMM | (c) Discrete RepL |

Figure 2: Empirical structure of (a) latent Dirichlet allocation model (in plate notation), (b) standard hidden Markov model, and (c) discrete representation learning.

## 4.1 Latent Dirichlet Allocation

Let us consider a corpus $\mathcal{D}$ of $M$ independent documents where each document is a sequence of $N$ words denoted by $W = (W_1, W_2, \cdots, W_N)$. Documents are represented as random mixtures over $K$ latent topics, each of which is characterized by a distribution over words. Let $V$ be the size of a vocabulary indexed by $\{1, \cdots, V\}$. Latent Dirichlet Allocation (LDA) [12] dictates the following generative process for every document in the corpus:

---

[1]Our code is anonymously published at https://anonymous.4open.science/r/OTP-7944/.

1. Sample $\theta \sim \mathrm{Dir}(\alpha)$ with $\alpha < 1$,
2. Sample $\gamma_k \sim \mathrm{Dir}(\beta)$ where $k \in \{1, \cdots, K\}$,
3. For each of the word positions $n \in \{1, \cdots, N\}$,

  - Sample a topic $Z_n \sim \mathrm{Multi\text{-}nominal}(\theta)$,
  - Sample a word $W_n \sim \mathrm{Multi\text{-}nominal}(\gamma_k)$,

where $\mathrm{Dir}(.)$ is a Dirichlet distribution. $\theta$ is a $K-$dimensional vector that lies in the $(K-1)-$simplex and $\gamma_k$ is a $V-$dimensional vector represents the word distribution corresponding to topic $k$. In the standard model, $\alpha, \beta, K$ are hyper-parameters and $\theta, \gamma$ are learnable parameters. Throughout the experiments, the number of topics $K$ is assumed known and fixed.

**Parameter Estimation.** To test whether OTP-DAG can recover the true parameters, we generate synthetic data in a simplified setting: the word probabilities are parameterized by a $K \times V$ matrix $\gamma$ where $\gamma_{kn} := P(W_n = 1 | Z_n = 1)$; $\gamma$ is now a fixed quantity to be estimated. We set $\alpha = 1/K$ uniformly and generate small datasets for different number of topics $K$ and sample size $N$. Inspired by the setup of [20], for every topic $k$, the word distribution $\gamma_k$ can be represented as a square grid where each cell, corresponding to a word, is assigned an integer value of either $0$ and $1$, indicating whether a certain word is allocated to the $k^{th}$ topic or not. As a result, each topic is associated with a specific pattern. For simplicity, we represent topics using horizontal or vertical patterns (See Figure 3). Following the above generative model, we sample 3 sets of data w.r.t 3 sets of configuration triplets $\{K, M, N\}$: $\{10, 1000, 100\}$, $\{20, 5000, 200\}$ and $\{30, 10000, 300\}$.

We compare OTP-DAG with Batch EM [38] and SVI [25, 27]. For the baselines, only $\gamma$ is learnable whereas $\alpha$ is set fixed to be uniform, whereas for our method OTP-DAG, we take on a more challenging task of **learning both parameters**. We report the fidelity of the estimation of $\gamma$ in Table 1 wherein OTP-DAG is shown to yield estimates closest to the ground-truth values. At the same time, our estimates for $\alpha$ (averaged over $K$) are nearly $100\%$ faithful at $0.10, 0.049, 0.033$ (recall that the ground-truth $\alpha$ is uniform over $K$ where $K = 10, 20, 30$ respectively).

Figure 3 illustrates the model topic distribution at the end of training. OTP-DAG recovers all of the ground-truth patterns, and as further shown Figure 4, most of the patterns in fact converge well before training ends.

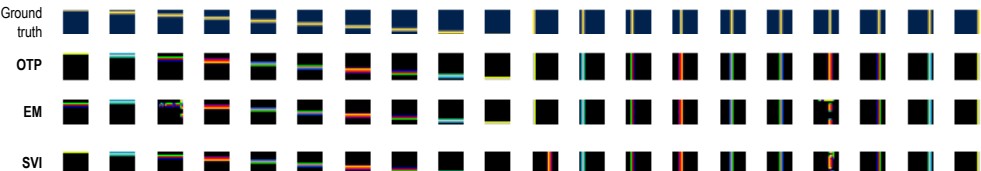

Figure 3: The topic-word distributions recovered from each method after $300-$epoch training. A grid corresponds to the word distribution of a topic. We use horizontal and vertical patterns in different colors to distinguish topics from one another. OTP-DAG recovers all ground-truth patterns.

**Topic Evaluation.** In this application, we use OTP-DAG to infer the topics of 3 real-world datasets:[2] 20 News Group, BBC News and DBLP. We here revert to the original generative process where the topic-word distribution follows a Dirichlet distribution parameterized by the concentration parameters $\beta$, instead of having $\gamma$ as a fixed quantity. $\beta$ is now initialized as a matrix of real values $\left(\beta \in \mathbb{R}^{K \times V}\right)$ representing the log concentration values. Table 2 reports the quality of the inferred topics from OTP-DAG, in comparison with Batch EM and SVI. For every topic $k$, we select top 10 most related words according to $\gamma_k$ to represent it. Topic quality is evaluated via the diversity and coherence of the selected words. Diversity refers to the proportion of unique words, whereas Coherence is measured with normalized pointwise mutual information [2], reflecting the extent to which the words in a topic are associated with a common theme.

---

[2] https://github.com/MIND-Lab/OCTIS.

Table 1: Fidelity of estimates of the topic-word distribution $\gamma$ across 3 settings. Fidelity is measured via $\mathbb{KL}$, $\mathbb{JS}$ divergence and Hellinger ($\mathbb{HL}$) distance [22] with the ground-truth distributions.

| Metric | $K$ | $M$ | $N$ | OTP-DAG (Ours) | Batch EM | SVI |
|---|---|---|---|---|---|---|
| $\mathbb{KL}\downarrow$ | 10 | 1,000 | 100 | **0.90 ± 0.14** | 1.61 ± 0.02 | 1.52 ± 0.12 |
| $\mathbb{JS}\downarrow$ | 10 | 1,000 | 100 | **0.68 ± 0.04** | 0.98 ± 0.06 | 0.97 ± 0.09 |
| $\mathbb{HL}\downarrow$ | 10 | 1,000 | 100 | **2.61 ± 0.08** | 2.69 ± 0.03 | 2.71 ± 0.09 |
| $\mathbb{KL}\downarrow$ | 20 | 5,000 | 200 | **1.29 ± 0.23** | 2.31 ± 0.11 | 2.28 ± 0.04 |
| $\mathbb{JS}\downarrow$ | 20 | 5,000 | 200 | **1.49 ± 0.12** | 1.63 ± 0.06 | 1.61 ± 0.03 |
| $\mathbb{HL}\downarrow$ | 20 | 5,000 | 200 | **3.91 ± 0.03** | 4.26 ± 0.08 | 4.26 ± 0.10 |
| $\mathbb{KL}\downarrow$ | 30 | 10,000 | 300 | **1.63 ± 0.01** | 2.69 ± 0.07 | 2.66 ± 0.11 |
| $\mathbb{JS}\downarrow$ | 30 | 10,000 | 300 | **1.53 ± 0.01** | 2.03 ± 0.04 | 2.02 ± 0.07 |
| $\mathbb{HL}\downarrow$ | 30 | 10,000 | 300 | **4.98 ± 0.02** | 5.26 ± 0.08 | 5.21 ± 0.09 |

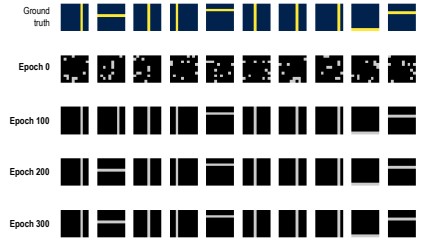

Figure 4: Converging patterns of 10 random topics from our OTP-DAG after $100, 200, 300$ iterations.

Table 2: Coherence and Diversity of the inferred topics for the 3 real-world datasets ($K = 10$)

| Metric | OTP-DAG (Ours) | Batch EM | SVI |
|---|---|---|---|
| | 20 News Group | | |
| Coherence (%) ↑ | **7.98 ± 0.69** | 6.71 ± 0.16 | 5.90 ± 0.51 |
| Diversity (%) ↑ | 75.33 ± 2.08 | 72.33 ± 1.15 | **85.33 ± 5.51** |
| | BBC News | | |
| Coherence (%) ↑ | **9.79 ± 0.58** | 8.67 ± 0.62 | 7.84 ± 0.49 |
| Diversity (%) ↑ | 86.00 ± 2.89 | 86.00 ± 1.00 | **91.00 ± 2.31** |
| | DBLP | | |
| Coherence (%) ↑ | 3.90 ± 0.76 | **4.52 ± 0.53** | 1.47 ± 0.39 |
| Diversity (%) ↑ | 84.67 ± 3.51 | 81.33 ± 1.15 | **92.67 ± 2.52** |

## 4.2 Hidden Markov Models

**Poisson Time-series Data Segmentation.** This application deals with time-series data following a Poisson hidden Markov model (See Figure 2b). Given a time series of $T$ steps, the task is to segment the data stream into $K$ different states, each of which is associated with a Poisson observation model with rate $\lambda_k$. The observation at each step $t$ is given as

$$P(X_t|Z_t = k) = \text{Poi}(X_t|\lambda_k), \quad \text{for } k = 1, \cdots, K.$$

Following [39], we use a uniform prior over the initial state. The Markov chain stays in the current state with probability $p$ and otherwise transitions to one of the other $K - 1$ states uniformly at random. The transition distribution is given as

$$Z_1 \sim \text{Cat}\left(\left\{\frac{1}{4}, \frac{1}{4}, \frac{1}{4}, \frac{1}{4}\right\}\right), \quad Z_t|Z_{t-1} \sim \text{Cat}\left(\left\{ \begin{array}{ll} p & \text{if } Z_t = Z_{t-1} \\ \frac{1-p}{4-1} & \text{otherwise} \end{array} \right\}\right)$$

Let $P(Z_1)$ and $P(Z_t|Z_{t-1})$ respectively denote these prior transition distributions. We generate a synthetic dataset $\mathcal{D}$ of 200 observations at rates $\lambda = \{12, 87, 60, 33\}$ with change points occurring at times $(40, 60, 55)$. We would like to learn the concentration parameters $\lambda_{1:K} = [\lambda_k]_{k=1}^K$ through which segmentation can be realized, assuming that the number of states $K = 4$ is known.

Table 3: Estimates of $\lambda_{1:4}$ at various transition probabilities $p$ and $L_1$ distance to the true values.

| $p$ | $\lambda_1 = 12$ | $\lambda_2 = 87$ | $\lambda_3 = 60$ | $\lambda_4 = 33$ | $\lambda_1 = 12$ | $\lambda_2 = 87$ | $\lambda_3 = 60$ | $\lambda_4 = 33$ |
|---|---|---|---|---|---|---|---|---|
| | **OTP-DAG Estimates** (Ours) | | | | **MAP Estimates** | | | |
| 0.05 | 11.83 | 87.20 | 60.61 | 33.40 | 14.88 | 85.22 | 71.42 | 40.39 |
| 0.15 | 11.62 | 87.04 | 59.69 | 32.85 | 12.31 | 87.11 | 61.86 | 33.90 |
| 0.35 | 11.77 | 86.76 | 60.01 | 33.26 | 12.08 | 87.28 | 60.44 | 33.17 |
| 0.55 | 11.76 | 86.98 | 60.15 | 33.38 | 12.05 | 87.12 | 60.12 | 33.01 |
| 0.75 | 11.63 | 86.46 | 60.04 | 33.57 | 12.05 | 86.96 | 59.98 | 32.94 |
| 0.95 | 11.57 | 86.92 | 60.36 | 33.06 | 12.05 | 86.92 | 59.94 | 32.93 |
| $L_1 \downarrow$ | **0.30** | **0.19** | **0.25** | **0.30** | 0.57 | 0.40 | 2.32 | 1.43 |

Table 3 demonstrates the quality of our estimates, in comparison with MAP estimates. Our estimation approaches the ground-truth values comparably to MAP. We note that the MAP solution requires the analytical marginal likelihood of the model, which is not necessary for our method. Fig-

ure [5a] reports the most probable state for each observation, inferred from our backward distribution $\phi(X_{1:T})$. It can be seen that the partition overall aligns with the true generative process the data.

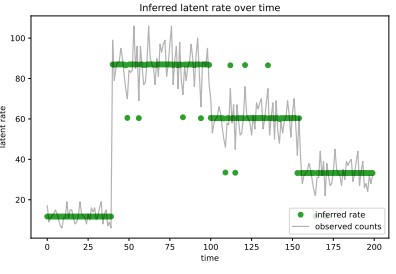

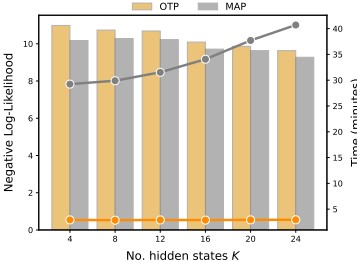

(a) Poisson time-series segmentation

(b) Polyphonic music modeling

Figure 5: (a) Segmentation of Poisson time series inferred from the backward distribution $\phi(X_{1:T})$. (b) Training time ↓ (in minutes) and Negative log-likelihood ↓ on the test dataset at various $K$.

**Polyphonic Music Modeling.** We consider another application of HMM to model sequences of polyphonic music. The data under analysis is the corpus of $382$ harmonized chorales by J. S. Bach [3]. The training set consists of $N = 229$ sequences, each of which has a maximum length of $T = 129$ and $D = 51$ notes. The data matrix is a Boolean tensor of size $N \times T \times D$. We follow the standard preprocessing where 37 redundant notes are dropped.[3]

The observation at each time step is modeled using a factored observation distribution of the form

$$P(X_t|Z_t = k) = \prod_{d=1}^{D} \text{Ber}(X_{td}|B_d(k)),$$

where $B_d(k) = P(X_{td} = 1|Z_t = k)$ and $k = 1, \cdots, K$. Similarly, we use a uniform prior over the initial state. Following [39], the transition probabilities are sampled from a Dirichlet distribution with concentration parameters $\alpha_{1:K}$, where $\alpha_k = 1$ if the state remains and $0.1$ otherwise,

$$Z_1 \sim \text{Cat}(\{1/K\}), \quad Z_t|Z_{t-1} \sim \text{Cat}(p), \quad p \sim \text{Dir}\left(\left\{\begin{array}{ll} 1.0 & \text{if } Z_t = Z_{t-1} \\ 0.1 & \text{otherwise} \end{array}\right\}\right).$$

The parameter set $\theta$ is a matrix size $D \times K$ where each element $\theta_{ij} \in [0, 1]$ parameterizes $B_d k(.)$. The goal is to learn these probabilities with underlying HMM sharing the same structure as Figure [2b]. The main difference is that the previous application only deals with one sequence, while here we consider a batch of sequences. For larger datasets, estimating MAP of an HMM can be expensive. Figure [5b] reports negative log-likelihood of the learned models on the test set, along with training time (in minutes) at different values of $K$. Our fitted HMM closely approaches the level of performance of MAP. Both models are optimized using mini-batch gradient descent, yet OTP-DAG runs in constant time (approx. 3 minutes), significantly faster than solving MAP with SGD.

### 4.3 Learning Discrete Representations

Many types of data exist in the form of discrete symbols e.g., words in texts, or pixels in images. This motivates the need to explore the latent discrete representations of the data, which can be useful for planning and symbolic reasoning tasks. Viewing discrete representation learning as a parameter learning problem, we endow it with a probabilistic generative process as illustrated in Figure [2c]. The problem deals with a latent space $\mathcal{C} \in \mathbb{R}^{K \times D}$ composed of $K$ discrete latent sub-spaces of $D$ dimensionality. The probability a data point belongs to a discrete sub-space $c \in \{1, \cdots, K\}$ follows a $K-$way categorical distribution $\pi = [\pi_1, \cdots, \pi_K]$. In the language of VQ-VAE, each $c$ is referred to as a *codeword* and the set of codewords is called a *codebook*. Let $Z \in \mathbb{R}^D$ denote the latent variable in a sub-space. On each sub-space, we impose a Gaussian distribution parameterized by $\mu_c, \Sigma_c$ where $\Sigma_c$ is diagonal. The data generative process is described as follows:

1. Sample $c \sim \text{Cat}(\pi)$,
2. Sample $Z \sim \mathcal{N}(\mu_c, \Sigma_c)$

---

[3]https://pyro.ai/examples/hmm.html.

3. Quantize $\mu_c = Q(Z)$,

4. $X = \psi_\theta(Z, \mu_c)$.

where $\psi$ is a highly non-convex function with unknown parameters $\theta$ and often parameterized with a deep neural network. $Q$ refers to the quantization of $Z$ to $\mu_c$ defined as $\mu_c = Q(Z)$ where $c = \operatorname{argmin}_c d_z(Z; \mu_c)$ and $d_z = \sqrt{(Z - \mu_c)^T \Sigma_c^{-1} (Z - \mu_c)}$ is the Mahalanobis distance.

The goal is to learn the parameter set $\{\pi, \mu, \Sigma, \theta\}$ with $\mu = [\mu_k]_{k=1}^K, \Sigma = [\Sigma_k]_{k=1}^K$ such that the model captures the key properties of the data. Fitting OTP-DAG to the observed data requires constructing a backward map $\phi : \mathcal{X} \mapsto \mathbb{R}^D$ from the input space back to the latent space. In connection with vector quantization, the backward map is defined via $Q$ and an encoder $f_e$ as

$$\phi(X) = \big[f_e(X), Q(f_e(X))\big], \quad Z = f_e(X), \quad \mu_c = Q(Z).$$

Following VQ-VAE [52], our practical implementation considers $Z$ as an $M-$component latent embedding. We experiment with images in this application and compare OTP-DAG with VQ-VAE on 3 popular datasets: CIFAR10, MNIST and SVHN. Since the true parameters are unknown, we assess how well the latent space characterizes the input data through the quality of the reconstruction of the original images. Our analysis considers various metrics measuring the difference/similarity between the two images on patch (SSIM), pixel (PSNR), feature (LPIPS) and dataset (FID) levels. We also compute Perplexity to evaluate the degree to which the latent representations $Z$ spread uniformly over $K$ sub-spaces. Table 4 reports our superior performance in preserving high-quality information of the input images. VQ-VAE suffers from poorer performance mainly due to an issue called *codebook collapse* [59] where most of latent vectors are quantized to few discrete codewords, while the others are left vacant. Meanwhile, our framework allows for control over the number of latent representations assigned to each codeword through learning $\pi$, ensuring all codewords are utilized. See Appendix C.3 for detailed formulation and qualitative examples.

Table 4: Quality of the image reconstructions ($K = 512$).

| Dataset | Method | Latent Size | SSIM ↑ | PSNR ↑ | LPIPS ↓ | rFID ↓ | Perplexity ↑ |
|---|---|---|---|---|---|---|---|
| CIFAR10 | **VQ-VAE** | $8 \times 8$ | 0.70 | 23.14 | 0.35 | 77.3 | 69.8 |
| | **OTP-DAG** (Ours) | $8 \times 8$ | **0.80** | **25.40** | **0.23** | **56.5** | **498.6** |
| MNIST | **VQ-VAE** | $8 \times 8$ | **0.98** | 33.37 | 0.02 | 4.8 | 47.2 |
| | **OTP-DAG** (Ours) | $8 \times 8$ | **0.98** | **33.62** | **0.01** | **3.3** | **474.6** |
| SVHN | **VQ-VAE** | $8 \times 8$ | 0.88 | 26.94 | 0.17 | 38.5 | 114.6 |
| | **OTP-DAG** (Ours) | $8 \times 8$ | **0.94** | **32.56** | **0.08** | **25.2** | **462.8** |

## 5   Limitations

Our framework employs amortized optimization that requires continuous relaxation or reparameterization of the underlying model distribution to ensure the gradients can be back-propagated effectively. For discrete distributions and for some continuous ones (e.g., Gamma distribution), this is not easy to attain. To this end, a recent proposal on *Generalized Reparameterization Gradient* [46] is a viable solution. OTP-DAG also relies on the expressivity of the backward maps. Since our backward mapping only considers local dependencies, it is however simpler to find a good approximation compared to VI where the variational approximator should ideally characterize the entire global dependencies in the graph. We use neural networks to model the backward conditionals. With enough data, network complexity, and training time, the difference between the modeled distribution and the true conditional can be assumed to be smaller than an arbitrary constant $\epsilon$ based on the universal approximation theorem [28].

## 6   Conclusion and Future Work

This paper contributes a novel approach based on optimal transport to learning parameters of directed graphical models. The proposed algorithm OTP-DAG is general and applicable to any directed graph with latent variables regardless of variable types and structural dependencies. As for future research, this new perspective opens up promising avenues, for instance applying OTP-DAG to structural learning problems where edge existence and directionality can be parameterized for continuous optimization, or extending it to learning undirected graphical models.

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
