# OpenReview forum: "Learning Directed Graphical Models with Optimal Transport"
_NeurIPS.cc/2023/Conference — Submitted to NeurIPS 2023_

### Official Review · Reviewer_HQH3 · 2023-07-04

**Soundness:** 3 good
**Presentation:** 2 fair
**Contribution:** 3 good
**Rating:** 5
**Confidence:** 3

**Summary:**

This paper uses optimal transport for learning the parameters of a DAG structure that represents a Bayesian network given samples drawn from it (data). In the proposed algorithm the data can be incomplete (i.e. some random variables may be latent).
The algorithm can be seen as a generalization of the existing "Optimal transport-based divergence minimization" on target distributions that are factorized as a product of conditional univariate densities.


**Strengths:**

1. The generalization of the Optimal Transport to Bayesian networks is interesting and as far as I can say novel (though, more or less straightforward).
2. The paper is well-written (In particular the first two sections).
3. Several experimental models are presented.


**Weaknesses:**

1. This method is only suitable for learning the parameters of Bayesian network DAGs. It can not be used for learning the DAG structure.
2. The description of the proposed algorithm (section 3 of the draft) is quite compact and many details are missing.
More details are presented in the supplementary material (Section B). Still, even there, the cost function and push-forward divergence measure that are used in the paper's experiments are not spelled out.




**Questions:**

I would suggest moving section B of the supplementary material to the main text and moving the details of the experimental models to the supplementary material.

**Limitations:**

yes

---

> ### Author Rebuttal · Authors · 2023-08-08
>
> We thank the reviewer for acknowledging the novelty of our work. We address the reviewer's concerns as follows:
>
> **1. This method is only suitable for learning the parameters of Bayesian network DAGs. It can not be used for learning the DAG structure.**
>
> The primary goal of this work is to introduce a new approach to learning graphical models based on optimal transport, with parameter estimation (given the DAG structure) as the foundation stone. Parameter estimation in graphical models is a general, long-standing, and challenging problem in machine learning, with numerous specialized methods developed to address it. However, we note that our framework is sufficiently general to serve as a powerful backbone to tackle other graph structures and learning problems.
>
> Although we leave the discussion on structural learning for future works, as mentioned in Section 6, extending OTP-DAG for this task is feasible. A starting point is to parameterize the DAG structure with a learnable (weighted) adjacency matrix and consider it as part of the model parameters to be learned. However, one needs to deal with further constraints specific to structural learning tasks, including the acyclicity of the DAG and important model assumptions such as faithfulness or sufficiency. Handling these aspects altogether deserves a separate contribution.
>
> **2. The description of the proposed algorithm is quite compact and many details are missing. Still, even there, the cost function and push-forward divergence measure that are used in the paper's experiments are not spelled out.**
>
> We have reported the formulation and other experimental technicalities for each application in Appendix C, including the choices of cost function and push-forward divergence. We recap these details in the following:
>
> - In the topic modeling task (see Appendix C.1, page 5), we use the cross-entropy loss as the cost function and exact Wasserstein distance as the push-forward divergence measure.
>
> - For hidden Markov models (see Appendix C.2, page 7), we use the KL divergence to optimize the push-forward constraint. As for the cost metric, the smooth $L_1$ loss and cross-entropy loss functions are used respectively for the Poisson time-series data segmentation task and the Polyphonic music modeling task.
>
> - In the discrete representation learning task (see Appendix C.3, page 9), the cost function is chosen to be the mean squared error. The push forward constraints are realized in $3$ additional loss terms, for which we use a combination of Wasserstein distance and KL divergence.
>
>
> We hope such diversity in the experimental setup can demonstrate the versatility of our method. Subject to space constraints, we will attempt to bring some of these details to the main paper.

---

> > ### Comment · Area_Chair_gwfv · 2023-08-18
> > **Thanks**
> >
> > Thank you for this feedback authors. This will be taken into account.

---

### Official Review · Reviewer_UL8t · 2023-07-06

**Soundness:** 3 good
**Presentation:** 3 good
**Contribution:** 2 fair
**Rating:** 5
**Confidence:** 4

**Summary:**

In this submission, the authors propose an optimal transport-based method to infer the parameters of probabilistic directed graphical models from partial observations.
In particular, given a DAG associated with the target model, the proposed method reparameterizes the probability of a node conditioned on its parents by an encoder with external perturbation.
Accordingly, a stochastic decoder is applied to map each node to the conditional density of its parents.
The above two modules lead to a model with an auto-encoding architecture, which can be learned like a Wasserstein autoencoder (WAE), as shown in Eqs. (2, 3).
Experiments on the inference of LDA, HMM, and discrete representation models demonstrate the potentials of the proposed method.

--- After rebuttal ---

Thanks for the authors' efforts in the rebuttal phase. After reading other reviewers' comments, my main concern about this work is still its similarity to WAE, especially the theoretical part. Although the authors claimed that WAE can be viewed as a special case of this work, in my opinion, it is more likely that this work is a special case of WAE. I am satisfied with the other part of this work, so my final score is kept as "borderline accept".

**Strengths:**

1. The paper is well-written and easy to follow.

2. The proposed method is reasonable --- the objective function is based on the theoretical part of WAE, whose rationality has been guaranteed. In addition, the implementation of the proposed method is simple.

3. The authors consider various applications, demonstrating the universality of the proposed method.

4. The limitations of the proposed method are discussed, and the potential solutions are provided at the same time.

**Weaknesses:**

1. If my understanding is correct, Theorem 1 in this submission is a special case of Theorem 1 in the WAE work [a]. The final objective (Eq.(3)) approximates the Wasserstein distance by relaxing the constraint of phi_i to a regularizer, which is also similar to the strategy of WAE. The authors should discuss the connections and the differences between the proposed method (and theory) and that in [a].

[a] Tolstikhin, Ilya, Olivier Bousquet, Sylvain Gelly, and Bernhard Schoelkopf. "Wasserstein Auto-Encoders." In International Conference on Learning Representations. 2018.


2. As the authors mentioned, the proposed method leverages the amortization strategy to reparameterize the conditional distributions. Therefore, in the experimental part, the authors should consider some amortization methods as baselines, e.g., those in [b, c, d].

[b] Kim, Yoon, Sam Wiseman, Andrew Miller, David Sontag, and Alexander Rush. "Semi-amortized variational autoencoders." In International Conference on Machine Learning, pp. 2678-2687. PMLR, 2018.

[c] Agrawal, Abhinav, and Justin Domke. "Amortized variational inference for simple hierarchical models." Advances in Neural Information Processing Systems 34 (2021): 21388-21399.

[d] Huang, Chin-Wei, Shawn Tan, Alexandre Lacoste, and Aaron C. Courville. "Improving explorability in variational inference with annealed variational objectives." Advances in neural information processing systems 31 (2018).


3. The datasets used in the experimental part are relatively simple and small. Especially in the experiments of discrete representation learning, I wonder 1) whether the proposed method can deal with images with larger sizes, e.g., face images in CelebA, and 2) besides reconstruction, whether the proposed method can generate images with tolerable qualities.

**Questions:**

See the comments in the section on weaknesses.

To demonstrate the novelty of the proposed method, the authors should clarify the connections and differences between the proposed method and WAE.

**Limitations:**

The authors discussed the limitations of using the amortization strategy at the end of the submission.
Some potential solutions are proposed at the same time.

---

> ### Author Rebuttal · Authors · 2023-08-09
>
> We thank the reviewer for acknowledging the novelty of our work. We address the reviewer's concerns in the following.
>
> **1. Connection with WAE:** WAE can be viewed as an application of OTP-DAG on a simple graphical model with only $2$ (sets of) nodes: the observed node $X$ and latent variables $Z$ (See Figure 1 in Appendix B). In this case, the backward mapping $\phi$ and forward mapping $\psi$ respectively play the role of the encoder and decoder. Likewise, both functions are jointly learned by minimizing the reconstruction loss and the push-forward divergence reduces to the prior matching term where $P_Z$ is part of the model generative process. OTP-DAG remains applicable when there are more parameters and hidden variables interplaying in a more complex structure. In terms of theoretical contribution, we offer an alternative, yet more straightforward approach to proving Theorem 1. This is achieved through Gluing lemma, which conclusively demonstrates the equivalence between the OT objective and the minimization of the reconstruction loss of observed samples.
>
> **2. Amortization baselines**
>
> We investigated two amortized variational inference (VI) methods: (1) Auto-encoding VI for topic modelling [e] and (2) Semi-amortized VAE [b]. We have published the codes for this part in our anonymous repository.
>
> **2.1. Auto-encoding VI for LDA:** Prod LDA is the proposed method in [e] that replaces the mixture model in LDA with a product of experts. We ran Prod LDA on 2 tasks: parameter estimation and topic inference. We also considered Neural LDA, which is standard LDA within the same variational auto-encoding approach. Implementation of these models is provided in the OCTIS library (https://github.com/MIND-Lab/OCTIS). We also use OCTIS to standardize evaluation for all models on the topic inference task.
>
> ***Parameter Estimation:*** To ensure a fair comparison, we train our model again using the architecture of the encoder of Neural/Prod LDA for our backward map, while keeping the other setting the same as reported in Appendix C.1. Table 3 reports the fidelity of the estimates. Prod LDA clearly outperforms Neural LDA since it is an improvement of the latter. Our method is generally on par with Prod LDA and we achieve such level of performance using a single learning procedure and without fine-tuning the hyper-parameters. Note that Prod LDA directly optimizes the ELBO, implicitly minimizing the KL divergence between the data and model distribution. This explains why it outperforms significantly in terms of this particular measure. For tractability, Prod LDA introduces a closed-form optimization objective that is specific to the choice of prior distributions, which is however not a requirement for our method.
>
> Furthermore, although Prod LDA can achieve a better estimation of the parameters numerically, it often fails to recover the correct distributions of words to topics. We refer to Figure 2 in the attached PDF file for the qualitative evidence, where we illustrate topic-word distribution patterns of randomly selected topics from all models. While our method may exhibit some inconsistencies in recovering accurate word distributions for each topic, these discrepancies are comparatively less pronounced when compared to Prod LDA and Neural LDA. This observation indicates a certain level of robustness in our approach.
>
> ***Topic Inference:*** This task assesses the quality of the inferred topics on real-world datasets. Note that the computation of topic coherence score using normalized pointwise mutual information in OCTIS is different than in the baseline paper. Concurring with what was reported in this paper, Prod LDA is generally superior to Neural LDA. However, compared to our OTP-DAG, the topics from Prod LDA achieves better Diversity yet with a much higher cost of Coherence on this task. Table 4 in the PDF file provides the empirical evidence for this task.
>
> [e] A. Srivastava and C. Sutton. Autoencoding Variational Inference For Topic Models. ICLR'17.
>
> **2.2. Semi-amortized VAE for Parameter Estimation:** We now study the capability of recovering the true parameters of semi-amortized VAE (SA-VAE) in comparison with our OTP-DAG. Borrowing the first setting in the paper (See Appendix B.1 in [b]), we create a synthetic dataset from a generative model of discrete sequences according to the following oracle generative process:
>
> $$z \sim \mathcal{N}(0,\mathbf{I}), \quad h_{t} = \text{LSTM} (h_{t-1}, x_{t} ), \quad x_{t+1} \sim \text{softmax}(\text{MLP}([h_t, z]))$$
>
> We here assume the architecture of the oracle is known and the task is simply to learn the parameters. Table 5 reports how well the estimated parameters approximates the ground-truth in terms of $L_1$ and $L_2$  distances, along with negative log-likelihood loss ($NLL$) of the reconstructed samples from the generative model. We also report the performance of a randomly initialized SA-VAE model to highlight the effect of learning. This empirical evidence again substantiate our competitiveness with popular amortization methods, while exhibiting more desirable properties than VI-based methods as already discussed in the main paper.
>
> **2.3. Experiments on CelebA**
>
> We further conducted an additional experiment on the CelebA to showcase the capacity of DAG-OTP on large datasets. The quantitative results are reported in Table 2.  We here again show our approach achieves a remarkable improvement in the codebook utilization, surpassing the baseline VQ-VAE by a substantial margin. Our better rFID scores in particular indicate our reconstructed images have higher quality at the dataset level. In Figure 3 in the PDF file, we present the generated samples from the CelebA dataset using Image transformer [f] as the generative model. These samples demonstrate that the discrete representation from the our method can be effectively utilized for image generation with acceptable quality.
>
> [f] Parmar, Niki, et al. Image transformer. ICML'18.

---

> > ### Comment · Reviewer_UL8t · 2023-08-11
> >
> > I keep my score unchanged after reading the authors' rebuttal and other reviewers' comments.

---

> > > ### Author Response · Authors · 2023-08-20
> > > **Thank you!**
> > >
> > > We sincerely appreciate the reviewer's time to engage in this interesting discussion. We will update our paper to include these valuable insights.

---

### Official Review · Reviewer_s48C · 2023-07-06

**Soundness:** 3 good
**Presentation:** 2 fair
**Contribution:** 2 fair
**Rating:** 5
**Confidence:** 3

**Summary:**

The authors propose a method for learning parameters $\theta$ of a DAG within an optimal transport (OT) framework, minimizing the Wasserstein distance between the data distribution $P_d$ and the model distribution $P_\theta$ in $\theta$. The Kantorovich formulation of this problem is a minimization of an expected cost $c(X,Y)$ over all joint distributions on $X,Y$ such that marginally $X \sim P_d, Y\sim P_\theta$; this is implemented in practice by empirically drawing $X_i$ from the data, and then $PA_{X_i}$ conditionally on $X_i$ to satisfy $PA_{X_i} \mid X_i \sim P_\theta(\textrm{PA}_{X_i})$ through the use of stochastic “backward” mappings. This makes optimization tractable over the space of backward mappings $\phi$ and model parameters $\theta$, so long as the constraint above is relaxed to a regularization term.


**Strengths:**

* DAGs represent an extremely rich family of models, and the work also generalizes to settings with unobserved variables, making this method easy to apply in a variety of settings.
* Unlike variational methods, evidence bounds need not be computed; the proposed method is more “direct” in this sense.
* The final optimization objective is easy to compute and computationally cheap.
* The framing of the problem from the lens of OT is novel to my knowledge, and provides an interesting formulation for optimization.
* The most significant strength of the paper is the evaluation of the method on a rich test suite of interesting problems such as LDA and Poisson time series segmentation. Comparisons show that the proposed method outperforms existing methods such as Batch EM and SVI in a variety of scenarios and metrics.


**Weaknesses:**

* The method reuqires that the random variables be reparameterizable in the sense of the equation at line 122 (this equation should be numbered); this may limit the family of joint distributions that can be considered. Though the authors say on line 167 that discrete variables can be used, reparameterization is tricky in these cases (as the authors acknowledge in the limitations section).
* The backward maps $\phi_i$ are a confusing quantity (and potentially hard to fit); see the Questions section. This could be due to a lack of understanding of my part. As I understand it, however, it raises questions about the efficacy of the proposed method.
* The formality of the OT framing is appealing, yet this formality is ultimately dropped for a regularized analogue that does not solve exactly the same problem that is posed.
* Some notation is confusing; for example, does $PA_{X_i}$ include exogenous variable $U_i$? The discussion around line 120 and line 147 suggest so, but then notationally the equations at line 122 and 136 suggest otherwise.


**Questions:**

* Is the data distribution $P_d(X_O)$ (line 125) a mixture of point masses, or a continuous (but unknown) distribution? This might be worth mentioning explicitly. In the case of the former, it could motivate use of the Kantorovich formulation instead of the Monge formulation of the OT problem.
* Line 128 can be expressed more precisely, and means to define $\mathcal{P}(X \sim P_d, Y \sim P_\theta)$ as the set of joint distributions on $X, Y$ with $P_d$, $P_\theta$ as marginals.
* In line 137, is this meant to say “backward” as it’s a backward map? The # notation should be defined somewhere.
* The constraint set on $\phi_i$ requires the following: that for $X_i \sim P_d$, the random variable $\phi_i(X_i) \sim P_\theta(\textrm{PA}_{X_i},  U_i)$. This ensures that $(X_i, \phi_i(X_i))$ are a joint distribution that marginally follow the requirements of the Kantorovich formulation. Is this all correct so far?
* If so, the constraint on the backward maps seems potentially ill-posed: by construction the distribution $P_\theta(PA_{X_i},  U_i)$ does not depend on the *value* of $X_i$ (it does depend on $X_i$ structurally, as we must know what nodes are its parents). For any given $\theta$, we are able to directly sample from $P_\theta(PA_{X_i},  U_i)$ via ancestral sampling. It seems to be then that training a neural network decoder such that $\phi_i(X_i) \sim P_\theta(PA_{X_i},  U_i)$ could result in a collapse to the “prior” defined the model, in other words the decoder could learn to ignore the input $X_i$ and simply mimic ancestral samples from $ P_\theta(PA_{X_i},  U_i)$, which presumably is what the stochastic backward maps are trained with. In such a case, the optimization over the space $\mathcal{P}(X \sim P_d, Y \sim P_\theta)$ would only occur over a very restricted subset of joint distributions, namely those where $X, Y$ are independent or nearly so. This seems like a possibility to me that is not addressed in the paper.
* In this extreme case, the final optimization problem becomes very simplistic: the regularization term disappears; and one simply draws ancestral samples from the model $P_\theta$, computes the cost function with a draw from the data, and minimizes the result in $\theta$. For simple models (or low dimensional $\theta$) this still might perform well. To ensure the situation above is not occurring maybe the quantity $KL(P_d P_\theta \mid \mid  P_d \phi)$ could be evaluated as a metric, or included via an additional regularization term to ensure that the backward maps $\phi$ are indeed resulting in distributions with dependence. This quantity should be significantly different from zero in that case.


**Limitations:**

The settings where the work can be applied are discussed by the authors. A limitations section is included that proactively assesses some limitations of the proposed method.

---

> ### Author Rebuttal · Authors · 2023-08-10
>
> We thank the reviewers for acknowledging the richness of our experiments. Our responses to the reviewer's questions are as follows:
>
> *Part 1: Questions*
>
> **1. Is the data distribution $P_d(X_{O})$ a mixture of point masses, or a continuous (but unknown) distribution?**
>
> The data distribution is the empirical one over observed dataset. To develop the theories, we indeed depart from the Kantorovich formulation in Eq. (1) where we consider the set of all couplings $\mathcal{P}(X \sim P_d(X_O), Y \sim P_{\theta}(X_O))$ instead of the deterministic couplings as in the Monge formulation.
>
> **2. The constraint set on $\phi_i$ requires the following that for $X_i \sim P_d$, the random variable $\phi_i(X_i) \sim P_\theta(PA_{X_i}, U_i)$. This ensures that $(X_i, \phi_i(X_i))$ are a joint distribution that marginally follow the requirements of the Kantorovich formulation. Is this all correct so far?**
>
> $\phi_i(X_i)$ might contain hidden variables, while the couplings $\mathcal{P}(X \sim P_d(X_O), Y \sim P_{\theta}(X_O))$ only contain observed variables from data aspect $P_d(X_O)$ and model distribution aspect $P_{\theta}(X_O))$. Therefore, $(X_i, \phi_i(X_i))$ is not a coupling in $\mathcal{P}(X \sim P_d(X_O), Y \sim P_{\theta}(X_O))$.
>
> **3. About the backward mapping:**
>
> Elaborating on the previous questions, we here aim to minimize the Wasserstein distance between $P_d(X_O)$ and $P_{\theta}(X_O)$ where $P_d(X_O)$ is the empirical data distribution and $P_{\theta}(X_O)$ is the model distribution over set of observed nodes $O$. Therefore, the coupling $\mathcal{P}(X \sim P_d(X_O), Y \sim P_{\theta}(X_O))$ is the set of joint distributions of $(P_d, P_{\theta})$ defined over the observed set $O$ that marginally follow the requirements of the Kantorovich formulation.
>
> The constraint set on $\phi_i$ maps $X_i$ (an observed variable where $i \in O$) to the marginal distribution (defined by the model) over *all* of its parent nodes which can include both hidden and observed nodes. Therefore, the couple $(X_i,\phi_i(X_i))$ does not necessarily follow the marginal constraints. If the parent set include observed nodes, the constraint set should also involve push-forwarding to the data distribution over such nodes. The stochastic backward maps are then trained to respect the dependencies in the data between $X_i$ and its parents.
>
> It is worth noting that our optimization objective in Eq. (3) includes a reconstruction term where we aim to find each backward map $\phi_i$ such that we can reconstruct the observed input $X_i$ from the forward aka the "model" direction effectively. In this sense, $X_i$ and its parents must obey the dependencies induced by the model.
>
> In an extreme case where all variables in $PA_{X_i}$ are latent, to the best of our understanding, the reviewer refers to a situation that resembles posterior collapse in training variational auto-encoders (VAEs). In the standard setting of VAEs where the underlying graphical model only contain two nodes, the parent variables are equivalent to the latent variables $Z$. In this case, our objective can reduce to the objective of Wasserstein auto-encoders (WAE) [1]. While VAE forces $Q(Z|X=x)$ to match the prior $P_Z$ for all the different input examples $x \sim P_X$, WAE aims to match $P_Z$ with the continuous mixture $Q_Z := \int Q(Z | X)dP_X$, thus encouraging the latent representation of examples to be far away and preventing them from collapsing ($Q$ denotes the variational distribution). Our push-forward divergence term shares the same purpose and in a general case $D$ is chosen to be Wasserstein distance. This additionally allows us to capture desirable properties of the Wasserstein distance, specifically in avoiding mode collapse problem which is known to occur commonly to f-divergences such as KL or JS [2].
>
> **4. Does $PA_{X_i}$ include exogenous variable $U_i$?**:
>
> We apologize for the confusion. The mention for $PA_{X_i}$ to include exogenous variable $U_i$ applies notationally to the discussion at line 147 to simplify the notations. The discussion above this line considers $PA_{X_i}$ and $U_i$ separately.
>
> **5. Line 128 can be expressed more precisely, and means to define $\mathcal{P}(X \sim P_d, Y \sim P_{\theta})$ as the set of joint distribution $X,Y$ with $P_d, P_{\theta}$ as marginals.**
>
> Thank you for pointing this out. We will update it in the revised paper.
>
> **6. Line 137, is this meant to say “backward” as it’s a backward map?**:
>
> It is meant to be "forward" as in the notion of "push forward" w.r.t the operator #. We will attempt to add a background introduction about this operator in the main paper.
>
> *Part 2: Weaknesses*
>
> **7. Reparametrization constraint:**
>
> In terms of the requirement for reparametrization, related approaches experiences similar inflexibility, i.e., advanced VI-based ones e.g., reparametrized VI and amortized VI. The proposal by Ruiz et al. (2016) was shown to accommodate a wider class of variational distributions, which can be used in our framework straightforwardly to alleviate the above issue.
>
> **8. The formality of the OT framing is appealing, yet this formality is ultimately dropped for a regularized analogue that does not solve exactly the same problem that is posed.**
>
> Relaxing the push-forward constraints into a regularized objective allows us to solve the learning problem efficiently with amortized optimization and to utilize the universal approximation capability of deep neural networks. This is a common practice in deep learning and also in OT formulations, which are shown to work effectively in practice, namely [1].
>
> [1] Tolstikhin et al. Wasserstein autoencoders. ICLR'18.
>
> [2] Arjovsky et al. Wasserstein generative adversarial networks. ICML'17.

---

> > ### Comment · Reviewer_s48C · 2023-08-16
> > **Rebuttal reply**
> >
> > Thanks for the detailed response. I’m not sure the authors have addressed my main concern about the backward maps $\phi$, which I’ll try to restate more clearly here. Let’s assume for ease there are no hidden variables and so all variables are observed. Further, let’s assume that there are only two nodes $X_c$, a child, and $X_p$ its sole parent. $X_p$ itself has no parents and is drawn from some prior in the generative model. My main concern is essentially:
> >
> > It is possible to satisfy the constraint below line 136 trivially if the decoder/backward map $\phi$ learns to imitate the prior, ignoring the numeric value of the input $X_c$ given to it (which $\phi$ might do, depending on how it is trained). In this case, the remainder of the minimization to occur by optimizing eq. (3) essentially will use the model distribution $P_\theta(X_c, X_p)$ and data distribution $P_d(X_c, X_p)$ which are independent of each other. In this case, the family of distributions used is severely diminished compared to that suggested by eq. (1): rather than using all joint distributions which admit $P_d, P_\theta$ marginally, only the product of these two distributions is used. The problem would collapse to the following:
> >
> > 1) Draw data pairs $(X_c, X_p)$ from the dataset $P_d$.
> > 2) Draw $(X_c, X_p)$ independently from the model $P_\theta$ for current value of $\theta$.
> > 3) Evaluate some cost between these draws; update $\theta$.
> >
> > The procedure becomes somewhat akin to approximate Bayesian computation (ABC) methods, which simulate data from the model and compare to real data, except in ABC parameter values are sampled from a distribution while in this case $\theta$ would be updated by gradient steps. This procedure could still work well on some problems; but nevertheless it’s totally different from the optimal transport problem posed. So a primary question to the authors is:
> >
> > **In construction of the backward maps $\phi$, how do the authors ensure optimization occurs over a *rich* family of joint distributions as used in eq. (1)? In particular, how does the training procedure of $\phi$ prevent the situation described above from occuring?**

---

> > > ### Author Response · Authors · 2023-08-18
> > > **Our responses (1/2) - Push-forward Divergence**
> > >
> > > We thank the reviewer for an insightful question.
> > >
> > > Let us first examine the reviewer's example. If the graphical model contains such two nodes and $X_p$ is also **observed**, this means for every sample, we know precisely which value of $X_c$ corresponds to which value of $X_p$. In this case, the backward map is used to transport $P_d(X_c)$ to $P_d(X_p)$ (i.e., not the prior on $X_p$).
> > > The reconstruction of $X_c$ can be evaluated by sampling $X_p$ directly from the data distribution $P_d(X_p)$. Moreover, in this case, if we want to parameterize and learn $P_d(X_p)$, we can further define $P_\theta(X_p \mid U_p)$ where $U_p$ is an exogenous variable sampled from its prior. We then have two backward maps from $X_c$ and $X_p$ to their parents respectively.
> > >
> > > If $X_p$ is **hidden**, we define a backward map $\phi$ over $X_c$ such that $\phi\\#P_d(X_c) = P(X_p)$ where $P(X_p)$ is the prior over $X_p$. The process described by the reviewer does not entirely agree with our algorithm. Our training procedure in this case is precisely as follows:
> > >
> > > (i) Draw $X_c \sim P_d(X_c)$.
> > >
> > > (ii) Draw $X_p \sim \phi(X_p | X_c)$.
> > >
> > > (iii) Draw $\tilde{X_c} \sim P_{\theta}(X_c | X_p)$.
> > >
> > > (iv) Evaluate the costs according to Eq. (3); update $\theta$.
> > >
> > > Our cost function specifically minimizes two terms:
> > > * push-forward divergence $D[P_{\phi}, P(X_p)]$ where $D$ is an arbitrary divergence. We use the WS distance for $D$ in our experiments.
> > > * the reconstruction loss between $X_c$ and $\widetilde{X}_c$.
> > >
> > > We now share some thoughts why this should work as we expect. Given $X_p \sim P(X_p)$ (i.e., its prior distribution), as $\phi\\# P_d(X_c) = P_\phi = P(X_p)$, this $X_p$ is sampled from $\phi\\# P_d(X_c) = P_\phi$. This further means that $X_p \sim \phi(X_p | X_c)$ for some $X_c \sim P_d(X_c)$ and $\tilde{X_c} \sim P_\theta(. | X_p) $ is close to this $X_c$ due to minimizing the reconstruction term. Therefore, $\tilde{X}_c$ is learned to follow the data distribution $P_d(X_c)$.
> > >
> > > When the backward $\phi$ mimics the prior, to our best understanding, the reviewer in fact refers to the posterior collapse problem notoriously occurring to VAE. We here explain why VAE is prone to posterior collapse and how our OT-based objective mitigates this issue.
> > >
> > > **1. The push-forward divergence**
> > >
> > > In the case of two nodes $X_p$ and $X_c$ where $X_p$ is latent, our framework OTP-DAG reduces to WAE [1] and the backward map corresponds to the variational encoder. Both WAE and VAE objectives entail the prior matching term. However, the two formulations are different in nature.
> > >
> > > Let $Q$ denote the set of variational distributions. The VAE objective is given as
> > >
> > > $$\inf_{\phi(X_p | X_c) \in Q} \mathbb{E_{P(X_c)}} [ \text{KL}(\phi(X_p | X_c), P(X_p))] - \mathbb{E_{\phi(X_p | X_c)}} [\log P_{\theta}(X_c | X_p)]$$
> > >
> > > By minimizing the above KL divergence term,  VAE basically tries to match the prior $P(X_p)$ for all different examples drawn from $P_d(X_c)$. Under VAE objective, it is thus easier for $\phi$ to collapse into a distribution independent of $P_d(X_c)$, where specifically latent codes are close to each other and reconstructed samples are concentrated around only few values.
> > >
> > > Meanwhile, for OTP-DAG/WAE, the regularizes in fact penalizes the discrepancy between $P(X_p)$ and $P_{\phi} := \mathbb{E}_{P(X_c)}[\phi(X_p | X_c)] $, which can be optimized using GAN-based, MMD-based or Wasserstein distance. For every sample $X_c \sim P_d(X_c)$, it is not $\phi(X_p | X_c)$ specifically that matches the prior, which encourages it to maintain the dependency between the latent codes and the input. Therefore, it is more difficult for $\phi$ to mimic the prior and trivially satisfy the push-forward constraint.

---

> > > > ### Author Response · Authors · 2023-08-18
> > > > **Our responses (2/2) - Reconstruction Loss**
> > > >
> > > > **2. The reconstruction loss**
> > > >
> > > > At some point of training, we however agree that there is still possibility to land at $\phi$ that yields samples $X_p$ independent of input $X_c$. If this occurs, $\phi \\# \delta x^{(1)}_c = \phi \\# \delta x^{(2)}_c = P(X_p)$ for any points $x^{(1)}_c, x^{(2)}_c \sim P_d(X_c)$. This means $\text{supp}(\phi(.|x_c^{1})) = \text{supp}(\phi(.|x_c^{2})) = \text{supp}(P(X_p))$, so it would result in a very large reconstruction loss because it requires to map $\text{supp}(P(X_p))$ to various $x_c^{1}$ and $x_c^{2}$. Thus our reconstruction term would heavily penalizes this. In other words, this term explicitly encourages the model to search for $\theta$ that reconstruct better, thus preventing the model from converging to the backward $\phi$ that produces sub-optimal ancestral samples.
> > > >
> > > > Meanwhile, for VAE, if the family $Q$ contains all possible conditional distribution $\phi(X_p | X_c)$, its objective is essentially to maximize the marginal log-likelihood $\mathbb{E_{P(X_c)}}[\log P_{\theta}(X_c)]$, or minimize the KL divergence $\text{KL}(P_d, P_{\theta})$. It is shown in [2] that under posterior collapse, VAE produces poor reconstructions yet the loss can still decrease i.e achieve low negative log-likelihood scores and still able to assign high-probability to the training data.
> > > >
> > > > To conclude, it is such construction and optimization of the backward that prevents the described situation from occurring. We here search for $\phi$ within a family of measurable functions and in practice approximate it with deep neural networks. It comes down to empirical decisions to select the architecture sufficiently expressive to each application.
> > > >
> > > > *References*
> > > >
> > > > [1] Tolstikhin et al. Wasserstein autoencoders. ICLR'18.
> > > >
> > > > [2] Dai et al. The Usual Suspects? Reassessing Blame for VAE Posterior Collapse. ICML'20

---

> > > > > ### Comment · Reviewer_s48C · 2023-08-19
> > > > > **Thanks**
> > > > >
> > > > > Thanks to the authors for the detailed response. I think the comparison with VAE/WAE is valuable and adding this comparison to the formulation (or as a special case) can help readers. The point discussed above regarding the "possibility to land at $\phi$ that yields samples $X_p$ independent of input $X_c$" is an important one due to the OT formulation, and I hope it can be explored in-depth more experimentally. The discussion by the authors above is a good start, and helps explain why independence might be difficult to occur, but ideally it would be verified experimentally that one gets different push-forward distributions $\phi(X_c)$ depending on the numeric input $X_c$, and the scale of the reconstruction loss and regularizer compared to justify the intuition provided above, the role of the tuning parameter $\eta$ in this, etc. After reading all reviews and rebuttals I leave my score unchanged.

---

> > > > > > ### Author Response · Authors · 2023-08-20
> > > > > > **Thank you!**
> > > > > >
> > > > > > We entirely agree with the reviewer's comments. One empirical evidence we can conveniently present here to support the intuition is in the application of Poisson time-series segmentation. Figure 5a (lines 265-266) specifically illustrates the state/segment inferred at $p=0.95$ for each observation directly from our backward map, which shows an alignment between the inferred states and the true ones.
> > > > > >
> > > > > > In the table below, we further analyze the quality of estimates and $L_1$ reconstruction loss at different values of $\eta$ (all results are averaged over 5 initializations). As we increase $\eta$ i.e., allowing the divergence term to dominate, the reconstruction loss does increase; however the estimated parameters only vary slightly and remain close to the ground-truth.  The experimental results reported in Figure 1 & Table 1 in the attached pdf file yield the same observation. However, we here fit the model with $p = 0.05$, a much poorer choice of prior and the model only shows a significant degradation in estimation performance from $\eta = 2.0$.
> > > > > >
> > > > > > We therefore believe that our algorithm should regularize the backward map effectively trying to maintain the dependency between the inferred ancestral values and the observations in order to recover the true parameters $\theta$.
> > > > > >
> > > > > > | $\eta$ | $\lambda_1 = 12$ | $\lambda_2 = 87$ | $\lambda_3 = 60$ | $\lambda_4 = 33$ | $L_1$ $\downarrow$ |
> > > > > > |:------:|:----------------:|:----------------:|:----------------:|:----------------:|:------------------:|
> > > > > > |   0.1  |       11.57      |       86.92      |       60.36      |       33.06      |        7.33        |
> > > > > > |   0.8  |       11.58      |       86.44      |       59.18      |       32.65      |        7.45        |
> > > > > > |   1.0  |       11.53      |       87.05      |       60.45      |       32.64      |        7.40        |
> > > > > > |   5.0  |       11.46      |       86.59      |       59.41      |       32.02      |        8.44        |
> > > > > >
> > > > > > We sincerely appreciate the reviewer's time to engage in this interesting discussion. We will update our paper to include these valuable insights.

---

### Official Review · Reviewer_uiRc · 2023-07-27

**Soundness:** 3 good
**Presentation:** 3 good
**Contribution:** 3 good
**Rating:** 6
**Confidence:** 3

**Summary:**

The authors propose a framework for learning the parameters of directed graphical models based on the idea of fitting by selecting the parameter values that minimize the Wasserstein distance (WD) between the data and model distributions. They prove (Thm. 1) that these distances can be characterized as the result minimizing a cost functional over a family of constrained stochastic 'backwards mappings', which yields a solvable objective when the constraint is relaxed and a regularization term is added to the objective.

The approach is illustrated via a series of experiments, including 3 real-world datasets, in which the proposed OPT-DAG method outperforms various baselines across disparate tasks.

**Strengths:**

I'm am not particularly familiar with the literature on DAG learning, nor on learning with 'optimal transport' objectives, so the extent of the originality of this paper is impossible to gauge. Operating under the assumption that this is the first work to introduce an OPT objective in this context, I would feel confident in stating that this is a fairly significant innovation.

The presentation of the paper is, in my opinion, of a high standard (modulo one or two concerns that I will voice in the next section). The appendices provide a considerable depth of exposition of their procedure, and I am satisfied that the most pressing immediate concerns are addressed, either there or in the main body.

**Weaknesses:**

I believe that the experimental section as well as the discussion could be improved. Again, I am unaware of what constitutes the benchmarking standard for DAG learning methods, but it seems to me as though the following are not adequately addressed:

a) The authors mention that their goal is not to achieve state-of-the-art performance, but rather to demonstrate the inherent versatility of their method. Situations in which their method might not be feasible are alluded to in Section 5 (Limitations), but the discussion here is extremely terse. Do these situations pose problems for competing alternatives as well? The chosen examples strike me as being somewhat simple. Do the authors claim that these examples are roughly representative of DAG learning problems generally?

b) For the topic evaluation example (239-250) OPT-DAG is outperformed by the baselines in the 'diversity' metric in all three datasets, and outperformed on 'coherence' on the DBLP data. These results are reported with precisely no discussion or explanation.

c) Section 4.2, first part (251- 266) could use some clarification. For example, the authors say "we generate a synthetic dataset D with 200 observations at rates {12, 87, 60, 33} with change points occurring at times (40, 60, 55)." If the model is as they say, (with a change of state happening with probability $1-p$, of which the lowest value is when $p = 0.95$, then a) why are there so few change points? At $p = 0.95$ would we not expect 10? Perhaps I am confused about what the authors are doing. Have they fixed the dataset, and computed the estimates of the rate parameters assuming that $p$ is fixed at the values indicated in the table (i.e. they fit the model 6 times, each time with a different assumed p)? Also, averaged over just the values $p = 0.75, 0.95$, OPT-DAG is actually inferior to MAP. Again, no explanation or discussion is forthcoming.

d) Generally, it is not clear to me if the baselines being compared against are the most appropriate. For instance, in the final example, VQ-VAE is used as a baseline, and its poor performance is linked to a phenomenon called 'codebook collapse'. The paper provided as a reference in fact proposes an extension to the vanilla version (of VQ-VAE) that they seem to be comparing against, which seems to indicate that not only is their baseline not state-of-the-art, it is in fact very well known to not be so...


**Questions:**

See above.

**Limitations:**

Limitations are discussed, but as per my comment above, it does not seem to me as though this discussion is adequate. The authors seem to have a variety of situations in mind in which their method will either not work or not be competitive, and a more explicit discussion here would be welcome.

---

> ### Author Rebuttal · Authors · 2023-08-08
>
> We thank the reviewers for acknowledging the originality of our proposed method. We respond to the reviewer's questions as follows:
>
> **1. Limitations:**
>
> In terms of the requirement for reparametrization, related approaches experiences similar inflexibility, i.e., advanced VI-based ones e.g., reparametrized VI and amortized VI. The proposal by Ruiz et al. (2016) was shown to accommodate a wider class of variational distributions, which can be used in our framework straightforwardly to alleviate the above issue.
>
> **2. Simplicity of applications:**
>
> Our paper focuses on the development of a fundamental and general framework for learning parameters for DAGs with latent variables. As agreed by Reviewer s48C, our evaluation has been conducted on *a rich test suite of interesting problems*, more concretely a wide range of models with different types of latent variables (continuous and discrete) and for different types of data (texts, images, and time series). Some of them might look simple, but all of them are fundamentally important, widely used, yet still challenging.  For example, Learning HMMs remains fairly challenging, with known optimization/inference algorithms (e.g., Baum-Welch algorithm) often too computationally costly to be used in practice. For learning discrete representations, despite the simple graph, the true generative function is unknown and it is often approximated with a deep neural network with a number of parameters that can scale up to millions. Solving those problems with EM or MAP would be both expensive and generally intractable.
>
> We believe that the models used in our experiments are representative and demonstrate our frameworks' applicability well. With the versatility of our framework, it has great potential to be applied to more complex models.
>
> **3. Topic Evaluation Results:**
>
> Diversity and Coherence metrics are mentioned in lines 244-248. These are popular metrics in assessing the performance of topic models in the unsupervised setting. There exists a trade-off between Diversity and Coherence: words that are excessively diverse greatly reduce coherence, while a set of many duplicated words yields higher coherence yet harms diversity. A well-performing topic model would strike a good balance between these metrics.  If we consider the two metrics comprehensively, our method achieves comparable or better performance than other learning algorithms. The datasets used for topic modeling in our experiments are quite diverse in terms of average document length : 20 News Group: 48, BBC News: 120, DBLP: 5 (all the numbers are rounded). The documents in DBLP are significantly shorter than the other two datasets and short texts are known to be challenging for both modeling and evaluation. Our method has comparable performance with others, but we agree that more study needs to be done to adapt our method to short texts better, which we leave to our future works.
>
> **4. Change point probabilities:**
>
> We first clarify the setting of time-series data segmentation. The HMM models a time series of integer counts associated with $K$ states, each of which follows a Poisson distribution of rate $\lambda_k$. The true transition probabilities are unknown and the value $p$ is treated as a hyper-parameter. The dataset is fixed and we fit HMM with 6 choices of $p$, for each of which we report the average results over 5 initialisations. Figure 5a reports the median of the most probable states inferred from 30 models at each step.
>
> The question now is how to choose $p$. Observing the data, one can assume $p$ to be relatively high, 0.75 - 0.95 seems most reasonable. This explains why MAP estimation at $p = 0.05$ is terrible. Meanwhile, for our OTP-DAG, the effect of $p$ is controlled by the trade-off coefficient $\eta$ (which regularizes the effect of the push-forward constraint). To make it comparable, in the experiments, we avoid tuning the parameters for our method and fix $\eta = 0.1$. The effect of $p$ on our performance is fairly minor, which explains OTP-DAG estimates across $p$ are less variant.  However, if we increase the $\eta$ weight and strongly force the model to fit $p = 0.05$, the model fits the data poorly and the performance degrades, in terms of both estimation and reconstruction quality. Table 1 in the PDF file provides empirical evidence for this claim, where we report OTP-DAG estimates and reconstruction losses at each $\eta$ value under the original settings. Figure 1 therein illustrates the predictions at each $\eta$, which also validates this claim.
>
> **5. Choice of baselines:**
>
> To clarify the rationale behind the chosen baselines, we must stress that our work focuses on parameter learning - that is to point estimate the model parameters given its known structure. Learning in the presence of latent variables often resorts to using EM or VI, which are fundamentally based on likelihood maximisation. Consequently, our experiments naturally entail a comparison with these approaches. In relation to VQ-VAE, the underlying technique is amortized inference, which is in fact a sub-class of VI. Note that discrete representation learning cannot be solved with EM or MAP. We are aware that VQ-VAE is not the state-of-the-art (SOTA), more importantly, nor do we aim to be one. The goal here is thus **not** to propose any SOTA model to discrete representation learning, rather to demonstrate the applicability of OTP-DAG to a problem that traditional methods such as EM, MAP or mean-field VI cannot simply tackle.  Yet motivated by the reviewer's suggestion, we additionally investigated a recent model called SQ-VAE [1] proposed to tackle the issue of codebook collapse. Table 2 in the PDF file reports the performance of SQ-VAE compared with our OTP-DAG on the same task where we again demonstrate our competitiveness with this SOTA model.
>
> [1] Takida et al. SQ-VAE: Variational Bayes on discrete representation with self-annealed stochastic quantization. ICML'22.

---

> > ### Comment · Reviewer_uiRc · 2023-08-17
> >
> > Thank you very much for your rebuttal. I am satisfied that my concerns have largely been addressed; after reading all the rebuttals, I leave my score unchanged.

---

> > > ### Author Response · Authors · 2023-08-20
> > > **Thank you!**
> > >
> > > We sincerely appreciate the reviewer's time to engage in this interesting discussion. We will update our paper to include these valuable insights.

---

### Author Rebuttal · Authors · 2023-08-10


We thank the reviewers for acknowledging the novelty of our method and richness of the experimentation. We immensely appreciate your support for acceptance of our paper. We here summarize the key points of our discussion with the reviewers.

Diverging from the existing approaches, we propose a new line of thinking to learning directed graphical models through the lens of optimal transport. Our method, OTP-DAG, is a versatile framework capable of addressing various problems with a single learning procedure that importantly can be automated. We demonstrate these merits across applications while maintaining competitive performance with prominent related approaches.

In this rebuttal, we reaffirm this message through supplementary clarifications and additional experimental investigations.

1. *To Reviewer uiRc and Reviewer HQH3:* we elaborate on our experimental results with further insights about the flexibility of our model through an ablation study.

2. *To Reviewer uiRc:* we investigated SQ-VAE, a recent model proposed to tackle codebook collapse issue; we show competitive quality of learned representations and reconstructed images with this SOTA model.

3. *To Reviewer UL8t:* we additionally compare our OTP-DAG with popular amortization baselines on parameter estimation and LDA topic modeling tasks; OTP-DAG competes on par with these models while bypassing the inconvenience of analytical derivation of the ELBO and its derivative.

4. *To Reviewer s48C:* we clarify our OT formulations, which sheds light on the merits of our method compared to competing approaches and the intuition behind our capability of recovering the true parameters that can respect the dependencies among variables in the graphical structure.

**Attached here is the PDF file including the figures and result tables from our experiments.**

---

### Decision · Program_Chairs · 2023-09-21

**Decision:**

Reject

**Comment:**

While the reviewers appreciated the motivation and clarity of the paper, they had concerns with (a) the experimental comparisons and discussion, (b) relationship to existing approaches, (c) the discussion on limitations. Specifically, for (a) the reviewers understood that the goal of the work was not to achieve state-of-the-art performance but to introduce a generic approach for learning parameters of directed graphical models, however, they argued that the authors should still compare with state-of-the-art methods in order to understand what the differences between OTP-DAG and SOTA is. In the submitted paper the authors compare with very outdated methods for topic modelling, HMM learning, and discrete latent representation learning. Against these OTP-DAG is superior, leading the reader to believe it could be a drop-in replacement for all problems. During the discussion the reviewers ask for more recent comparisons. Compared to more recent methods, OPT-DAG no longer clearly outperforms them, and is even clearly beaten across different settings. This raises a question: “If the method isn’t supposed to be state-of-the-art, and is outperformed by it in many experiments, when should it be used?” To answer this it would have been nice to see OTP-DAG used in a problem where generic approaches really are SOTA, as I suspect it would outperform these approaches. Without this, the motivation is much less strong. For (b), the reviewers were confused about the backward map and the relationship of the method to WAE. While there is a bit of description in the appendix, it is very brief. The authors made this relationship clearer in the discussion phase. The reviewers and I think the paper would be much stronger if it was framed as an extension of WAE-type methods. This would be much more educational for the community and it would make it clearer why this approach is useful. For (c), the reviewers were confused about the discussion in the limitations section. The authors did a good job clarifying one of the limitations (reparameterization) but did not clearly resolve the confusion of the reviewers on when the method may not be feasible, and whether the shortcomings also apply to alternative methods.
In fact, I found many reponses by the authors confusing, often responding to a point not directly raised by the reviewers. For instance, when a reviewer wanted clarification on the backward map, the authors point out that OTP-DAG simplifies to the WAE. This itself is interesting, but they have an extremely long and convoluted answer that includes the push-forward divergence (which the reviewer never asked about) and they go into further detail about the VAE objective (also never asked about)). In another case when a reviewer asks if Theorem 1 is a special case of Theorem 1 in the WAE the authors point out that they offer a more straightforward approach to proving Theorem 1, which is again not what the reviewer asked about. The authors also argue things in the discussion that are not clear or well-supported. For example they claim that they get similar results to Prod LDA but in fact Prod LDA clearly outperforms OTP-DAG in all cases. When talking about diversity and coherence they also claim that taken together they have comparable or better performance, but it is unclear what this means.
Given all of the above, I believe this work should be rejected at this time. Once these things and other issues mentioned in the reviews are addressed in an updated version, the work will be much improved.